# Balancing Regeneration and Resistance: Targeting DCLK1 to Mitigate Gastrointestinal Radiation Injury and Oncogenesis

**DOI:** 10.3390/cancers17122050

**Published:** 2025-06-19

**Authors:** Landon L. Moore, Jerry Jaboin, Milton L. Brown, Courtney W. Houchen

**Affiliations:** 1Department of Medicine, Digestive Diseases and Nutrition, The University of Oklahoma Health Sciences Center, Oklahoma City, OK 73104, USA; 2Department of Veterans Affairs Medical Center, Oklahoma City, OK 73104, USA; 3The Peggy and Charles Stephenson Cancer Center, Oklahoma City, OK 73104, USA; jerry-jaboin@ouhsc.edu; 4Department of Medicine, Macon and Joan Brock Virginia Health Sciences at Old Dominion University, Norfolk, VA 23501-1980, USA; brownml@odu.edu

**Keywords:** DCLK1, gastrointestinal acute radiation syndrome, ionizing radiation, oxidative stress, cancer stem cells, tumor microenvironment, radioresistance, radioprotection

## Abstract

Ionizing radiation, like that used in cancer treatments or from accidental or deliberate exposure, can harm the cells lining the gut, leading to serious digestive problems. A rare group of cells in the intestine—called tuft cells—produce a protein named DCLK1, which helps rebuild damaged tissue by activating repair and survival processes. However, the same abilities that protect healthy cells can also support the persistence of early cancer cells, making treatment-resistant tumors more likely to develop. This review explains how DCLK1 both aids healing and risks promoting cancer by influencing key protective and inflammatory pathways. We are exploring ways to measure DCLK1 levels as a marker of radiation damage and to design therapies that block its harmful effects—boosting gut recovery while reducing the chance of new cancers.

## 1. Introduction

Ionizing radiation (IR), including X-rays, gamma rays, neutrons, and alpha particles, possesses sufficient energy to ionize atoms and generate reactive oxygen species (ROS), resulting in DNA breaks, protein fragmentation, lipid peroxidation, and extensive cellular injury. Clinically, IR underpins cancer therapy but inflicts collateral damage to normal tissues and increases secondary malignancy risk [1,2,3]. High, acute doses precipitate acute radiation syndrome (ARS), with hematopoietic (H-ARS) and gastrointestinal (GI-ARS) subsyndromes driving early morbidity through bone marrow failure and intestinal barrier loss [4,5,6,7,8,9]. Fractionated radiotherapy delivers lower doses that cumulatively trigger similar pathologies, highlighting the tension between tumor eradication and tissue preservation [10,11,12]. Radioprotectors and mitigators, antioxidants, hematopoietic growth factors, and renin-angiotensin system (RAS) inhibitors improve survival when delivered within 24 h of exposure [13], yet they inadequately prevent late effects such as fibrosis, organ dysfunction, and carcinogenesis [14,15,16,17,18].

Recent studies identify a rare epithelial population—intestinal tuft cells expressing doublecortin-like kinase 1 (DCLK1)—as key mediators of post-IR regeneration [19,20]. DCLK1, a microtubule-associated serine/threonine kinase, was discovered through its homology to the neuronal doublecortin (DCX) family and is enriched in postnatal brain and gastrointestinal tissues [21,22,23,24]. It exists in multiple isoforms due to alternative promoter and splicing usage to generate in humans at least four distinct major isoforms (isoforms 1–4), which vary in microtubule-binding affinity and cellular localization (Figure 1). Structurally, all four isoforms possess a CaMK-like kinase domain and an autoinhibitory domain (AID) that regulates its phosphorylation activity [25]. Two variants (isoforms 1 and 2) are transcribed from the alpha promoter and encode proteins with tandem microtubule-binding (DCX) domains [26]. DCLK1 isoforms 2 and 4 additionally possess a unique extracellular, non-kinase C-terminal binding region (NKEBD) that is targeted by therapeutic agents such as the monoclonal antibody CBT-15 to disrupt tumor–stromal interactions in cancer [27,28,29]. Although its full substrate repertoire remains poorly defined—with only a handful of direct targets validated—DCLK1 modulates phosphorylation events that influence β-catenin, JNK, and downstream effectors in the MAPK and PI3K–AKT pathways, suggesting a broad role in cancer and tissue homeostasis.

DCLK1 is a marker for both tuft and cancer stem-like cells (CSCs) [30]. In the intestine, DCLK1^+^ tuft cells perform chemosensory, regenerative, and immunomodulatory roles, mirroring key functional traits of CSCs [31,32]. Quiescent under homeostasis, DCLK1^+^ tuft cells activate antioxidant defenses, DNA-repair pathways, and pro-regenerative cytokines following IR [33,34,35]. However, these same mechanisms—enhanced ROS neutralization, DNA repair, and pro-survival signaling via NF-κB, TGF-β, and Wnt/β-catenin—parallel CSC traits, heightening oncogenic risk during tissue regeneration [36]. Moreover, DCLK1^+^ cells modulate the tumor microenvironment (TME) by skewing immune polarization (ILC2, M2 macrophages, myeloid-derived suppressor cells (MDSCs), regulatory T cells) and remodeling the extracellular matrix (ECM) via fibroblast crosstalk [37,38]. This review integrates emerging insights into DCLK1’s dual roles in intestinal regeneration and oncogenesis under IR. We first summarize ARS pathophysiology and GI-ARS, then examine DCLK1-driven repair and intersecting signaling pathways (NF-κB, TGF-β, MAPK, p53). Finally, we evaluate DCLK1-targeted strategies—from small molecules to immunotherapies and nanomedicine—with the aim of enhancing radioprotection while mitigating malignancy risk.

## 2. DCLK1 Regulates Radiation Response in GI-ARS and Fractionated GI Injury

ARS arises when large doses of ionizing radiation are delivered over a short period, either accidentally or intentionally, overwhelming cellular repair mechanisms and precipitating multi-organ injury [9]. Clinically, ARS progresses through a prodromal phase—marked by nausea, vomiting and fatigue—followed by a transient latent period, and culminates in a manifest illness phase whose severity and timing depend on the absorbed dose [39] (Table 1). Three overlapping subsyndromes define ARS: H-ARS, GI-ARS, and central nervous system ARS (CNS-ARS). H-ARS develops at approximately 1–7 Gy and is characterized by bone-marrow aplasia, immunosuppression, and bleeding diatheses, often resulting in sepsis or hemorrhage without prompt supportive care [40,41,42,43]. GI-ARS emerges at doses of roughly 7–15 Gy, leading to crypt stem-cell loss, epithelial barrier breakdown, bacterial translocation, and a high risk of sepsis and multi-organ failure [44]. CNS-ARS, which follows exposures above 15 Gy, produces rapid neurovascular collapse and is almost universally fatal within hours [45,46]. Importantly, fractionated or chronic low-dose exposures—as encountered in radiotherapy—can recapitulate these syndromes through cumulative injury, underscoring the need to balance dose delivery schedules to optimize tumor control while preserving normal-tissue function [47]. Recent work shows that DCLK1-expressing tuft cells play a critical role in post-irradiation repair; in mice, a single 12 Gy total-body dose leads to robust Dclk1 upregulation in surviving crypts, activation of DNA-damage and antioxidant pathways, and improved epithelial regeneration—effects that mitigate GI-ARS severity [48]. Similarly, in vitro experiments using colorectal cancer lines reveal that silencing DCLK1 increases radiosensitivity at doses around 6 Gy, underscoring DCLK1’s function in DNA repair and cell survival signaling after moderate IR exposure [49].

### 2.1. Radiation Induced GI Damages

Radiation-induced GI damage results from direct cellular insults, e.g., DNA breaks, protein fragmentation, and lipid peroxidation. IR targets water molecules, leading to radiolysis and the rapid generation of reactive oxygen species (ROS) (Figure 2A). ROS inflicts additional damage—inducing DNA breaks (Figure 2B) [50], disrupting protein structure through fragmentation and crosslinking (Figure 2C) [51,52], and compromising membrane integrity by altering lipids (Figure 2D) [53]. IR amplifies ROS production in a dose-dependent manner [54]. For instance, ^137^Cs γ-ray exposure produces ~60 ROS per nanogram of tissue within a microsecond [55]. After a 6 Gy dose, ROS levels surge, decline over six hours, and normalize within two weeks, marking a crucial window for genotoxic and cytotoxic effects [56]. Physiologically, ROS serves as a critical regulator of cellular biochemistry [57]. Low ROS levels promote proliferation and differentiation, whereas high ROS concentrations lead to DNA damage, apoptosis, and necrosis [58,59]. However, excess ROS levels elevate mutation and cancer risk, drive inflammation, and disrupt the gut microbiome, thereby exacerbating gastrointestinal injury [60]. Preclinical models use single high doses, whereas clinical radiotherapy delivers multiple fractions. Fractionated irradiation (e.g., five daily stereotactic body irradiation fractions) alters crypt survival, cytokine profiles, and regenerative kinetics; palbociclib had protective effects under single-dose IR but worsened injury under fractionation [61]. Thus, clinically relevant fractionated models are essential for evaluating DCLK1-mediated repair and immunomodulation.

### 2.2. RAS Amplifies GI-IR Injury

Concurrently, the RAS amplifies oxidative stress and inflammation; IR-induced kidney damage increases renin, converting angiotensinogen to angiotensin I and then II (Ang II) via ACE (Figure 2E). Ang II engages AT_1_R on epithelial, endothelial, and fibroblast cells, activating NADPH oxidases and TGF-β signaling to drive ROS production, inflammation, vasoconstriction, and fibrosis [62,63,64,65,66]. Positive feedback loops further intensify injury by perpetuating ROS production and inflammatory signaling [67], while AT_2_R and ACE2/Ang (1–7)/Mas receptor axes counterbalance Ang II, IR-induced ROS disrupt this homeostasis [68,69,70]. Ang II stimulates TGF-β, a key mediator of fibrosis and intestinal remodeling [71], while angiotensin-(1–7) exerts an opposite effect via the Mas receptor and helps restore gut barrier integrity [72,73]. Preclinical studies indicate that RAS inhibitors, such as ACE inhibitors (ACEIs) and angiotensin receptor blockers (ARBs), reduce tissue damage and fibrosis in radiation-induced GI injury [74,75]. While endogenous antioxidants and anti-inflammatory defenses provide partial protection, persistent disruptions compromise GI epithelial regeneration.

### 2.3. Gastrointestinal Damages and DCLK1-Mediated Repair

A critical component of intestinal radiation response involves epithelial regeneration, primarily driven by DCLK1-positive tuft cells, which rapidly restore mucosal integrity following injury [33,34]. DCLK1 is mainly expressed in GI epithelia, especially tuft cells, and minimally in hematopoietic tissues. Although radiation impacts both hematopoietic and GI systems, DCLK1’s established role centers on intestinal repair [35], with recent evidence expanding its influence to immune modulation [30,76,77]. IR-induced GI injury varies with dose and anatomical location. The small intestine is highly susceptible to radiation, exhibiting villous atrophy and crypt depletion, resulting in malabsorption and compromised barrier function, which is in contrast to the colon that typically develops chronic inflammation, fibrosis, strictures, and bleeding [78,79,80]. Clinical radiation therapy (RT) protocols significantly differ in dose, fractionation, and targeted volumes, altering GI toxicity risks. Fractionated pelvic RT generally limits small bowel exposure beyond 45 Gy (V45), whereas hypofractionated stereotactic body radiotherapy (SBRT), with higher fractional doses (8–12 Gy), risks late toxicities if organ motion is inadequately controlled [81,82]. These clinical distinctions underscore the necessity for site-specific and dose-adaptive DCLK1-targeted interventions. Integrating therapeutic approaches targeting H-ARS and acute GI-ARS, while leveraging DCLK1 modulation for fractionated RT-induced GI injury, could effectively mitigate complex radiation-induced injuries.

Small intestinal epithelium, structured into crypt–villus units and the colon epithelium without villi, undergoes continuous renewal via a dynamic stem cell hierarchy [83]. Typically, crypt base cells proliferate, migrate upwards, and undergo apoptosis, rendering the epithelium acutely radiation-sensitive. Intestinal stem cells (ISCs), including actively cycling Lgr5^High^ crypt base columnar (CBC) cells and quiescent reserve ISCs, maintain epithelial homeostasis and repair radiation-induced damage [84,85]. Distinct from highly proliferative CBC cells, DCLK1-positive tuft cells are predominantly quiescent under homeostatic conditions but activate post-irradiation, promoting mucosal recovery [35,85]. Activation of these cells triggers signaling pathways, including NF-κB, TGF-β, and Wnt/β-catenin, enhancing cell survival and conferring therapeutic resistance. Thus, DCLK1 serves as a pivotal regulator, bridging mucosal repair processes with therapeutic resistance and potential oncogenicity.

## 3. DCLK1 in Radiation Response Pathways

In GI tissues, multiple epithelial cell types—including intestinal stem cells, tuft cells, goblet cells, and enteroendocrine cells—activate coordinated stress response pathways to counteract radiation-induced injury [86,87]. DCLK1, enriched in radiation-resistant tuft cells, engages directly or indirectly in a network of stress-adaptive signaling pathways that promote epithelial regeneration, immune modulation, and tumor resistance [30]. It intersects with autophagy through p62/SQSTM1 accumulation, with p62/SQSTM1 serving as a key adaptor protein linking oxidative stress to Nrf2 activation. DCLK1 also activates inflammatory circuits, including NF-κB and TGF-β, facilitating EMT and linking tissue repair to fibrosis and immune suppression. Through modulation of MAPK cascades (ERK, JNK, p38), DCLK1 promotes proliferation and survival, often in conjunction with p62-driven stress signaling. Moreover, DCLK1 influences the DNA damage response by enhancing p53-dependent repair while attenuating apoptosis, enabling the persistence of genetically unstable, therapy-resistant cells. Collectively, these pathways position DCLK1-positive tuft cells and CSCs as central mediators of post-irradiation regeneration and tumor radioresistance, establishing DCLK1 as both a key regulator of epithelial recovery and a potential oncogenic driver.

### 3.1. DCLK1, p62 and Autophagy

IR initially activates autophagy as a cytoprotective response; however, high-dose or prolonged exposure impairs this process [88,89]. Excessive ROS disrupt lysosomal integrity, inhibit autophagosome–lysosome fusion, and activate caspase-3, which cleaves Beclin-1 and suppresses autophagosome formation [90]. As a result, autophagy stalls, leading to p62/SQSTM1 accumulation and defective clearance of damaged cellular components [91]. In colon cancer models, autophagy inhibition causes p62 to accumulate and co-localize with DCLK1 in autophagosomes of murine and human tumors and is associated with preventing DCLK1 degradation and increased DCLK1 isoform promoter activity (Figure 3A) [92]. Furthermore, autophagy inhibition induces dose-dependent increases in both p62 and DCLK1 protein levels, along with enhanced DCLK1 transcriptional activation. Knockdown of essential autophagy genes (Beclin1 or Atg7) reduced DCLK1 promoter activity without altering protein stability, suggesting that p62 may stabilize DCLK1 post-transcriptionally. These findings support a model in which p62 sequesters DCLK1 isoforms, shielding them from degradation and promoting their accumulation in the tumor microenvironment—potentially enhancing stemness, immune evasion, and tumor progression. However, direct biochemical interaction between p62 and DCLK1 remains unproven.

Additionally, p62 accumulation sequesters Keap1, stabilizing Nrf2 and enhancing its transcriptional activity. This activation modulates mTOR signaling and supports radioresistance through selective autophagy [93,94]. Under basal conditions, Keap1 targets Nrf2 for proteasomal degradation [95]; oxidative stress disrupts this interaction, allowing for Nrf2 nuclear translocation and transcription of antioxidant response elements (AREs) [96,97]. Cancer cells exploit this axis by upregulating antioxidant enzymes such as SOD and catalase to resist IR-induced damage [98,99,100]. While no study to date directly links DCLK1 to Nrf2-driven radioresistance, its stabilization through p62 accumulation intersects multiple stress adaptation pathways and may facilitate oncogenic transformation.

### 3.2. DCLK1 and Inflammatory Responses

The NF-κB pathway regulates immune responses, inflammation, cell survival, and apoptosis. DNA double-strand breaks activate ATM kinase, which phosphorylates IKKγ and initiates NF-κB signaling (Figure 3B) [101,102]. Activated NF-κB promotes DNA repair, cell cycle progression, mitochondrial antioxidant responses, and cytokine expression, enhancing cell survival and, under chronic ROS exposure, facilitating tumor progression [103,104]. DCLK1 interacts with IKKβ, promoting its phosphorylation at Ser177/181 and activating NF-κB [105]. Indeed, DCLK1 silencing or inhibition blocks LPS-induced NF-κB activation and cytokine release, whereas DCLK1 upregulation increases NF-κB p65 activity, conferring resistance to apoptosis and enhancing DNA repair [106]. High DCLK1 expression also promotes an EMT-like phenotype by regulating pathways including NOTCH, WNT, receptor tyrosine kinases, and TGF-β [107].

A key driver of EMT, TGF-β contributes to radiation-induced fibrosis and chronic injury [108]. TGF-β signaling, enhanced by ROS, activates SMAD phosphorylation and nuclear translocation via NOX4 (Figure 3C, Canonical), promoting extracellular matrix (ECM) deposition, EMT, and fibrotic remodeling [109,110]. DCLK1 amplifies this cascade, upregulating EMT transcription factors (ZEB1, ZEB2, SNAIL, SLUG) and promoting mesenchymal traits linked to invasion, metastasis, and therapy resistance [111,112,113]. In irradiated tissues, DCLK1-driven EMT may exacerbate fibrosis and impair healing. Circulating DCLK1 and TGF-β levels also correlate with fibrosis progression in liver disease [114]. While feedback between DCLK1 and TGF-β ligands remains under study, current evidence supports a role for DCLK1 in potentiating TGF-β–mediated EMT. This interaction has dual consequences: (1) promoting therapeutic resistance in cancer by enriching stem-like, mesenchymal populations and (2) worsening fibrosis post-irradiation through maladaptive tissue remodeling. Through its integration with the TGF-β/EMT axis, DCLK1 links epithelial plasticity with fibrogenesis and metastasis.

### 3.3. DCLK1 and the MAPK Pathway

The MAPK pathway, including ERK, JNK, and p38 cascades, plays a critical role in mediating cellular responses to oxidative stress, DNA damage, and inflammation following ionizing radiation in the GI tract [115]. DCLK1 influences MAPK signaling in several notable ways. ROS activate TGF-β’s non-canonical MAPK cascades (ERK, JNK, and p38) (Figure 3C, Non-canonical) [116,117,118]. ERK1/2 activation promotes survival and proliferation, while JNK and p38 modulate apoptosis and inflammation [119,120,121]. Sustained JNK activation can promote apoptosis by phosphorylating Bax or enhancing c-Jun activity, while p38 induces inflammatory cytokines like TNF-α and interleukins [122,123]. Additionally, ROS oxidize thioredoxin to release inhibition of apoptosis signal-regulating kinase 1 (ASK1), which phosphorylates MAPKKs (MKK4/7 for JNK and MKK3/6 for p38), thereby activating transcription factors such as c-Jun and p53 [124,125,126]. In colorectal and pancreatic cancers, DCLK1 overexpression suppresses tumor-suppressive miR-143/145, indirectly enhancing KRAS expression and downstream ERK/MAPK activity [107]. p62 accumulation in DCLK1-positive cells further activates ERK and p38, establishing constitutive MAPK signaling that favors survival and proliferation, particularly after radiation-induced damage [92]. Sustained JNK activation can promote apoptosis by phosphorylating Bax or enhancing c-Jun activity, while p38 induces inflammatory cytokines like TNF-α and interleukins [122,123]. In radiation contexts, ERK1/2 facilitates cell survival and DNA repair, while p38 and JNK exert context-dependent effects on cell fate. Overall, DCLK1 promotes MAPK pathway activation via KRAS upregulation and p62 accumulation, thereby supporting tumor growth, stress adaptation, and potentially aiding intestinal epithelium regeneration after radiation injury.

### 3.4. DCLK1 and the DNA Damage (p53) Response

DCLK1 modulates p53 activity in a context-dependent manner. In quiescent gastrointestinal cells, DCLK1-positive tuft cells express 6–7-fold higher p53 mRNA levels than neighboring cells and overexpress Survivin, suggesting a state of enhanced genomic surveillance coupled with resistance to injury [127]. The p53 pathway orchestrates DNA damage responses by inducing cell cycle arrest or apoptosis (Figure 3D) [128]. Loss of p53 heightens sensitivity to radiation by increasing stem cell death and impairing epithelial repair [129,130,131,132], whereas its overexpression confers radioprotection [133,134]. IR activates ATM kinase, which stabilizes p53 to promote DNA repair or apoptosis in response to DNA lesions [135,136,137,138]. Beyond repair, p53 regulates redox homeostasis by inducing antioxidant (e.g., sestrins, TIGAR) or pro-oxidant (e.g., PIG3) genes depending on stress severity [139]. Thus, intact p53 function promotes epithelial regeneration, while its deficiency exacerbates GI toxicity.

In cancer or stress conditions, DCLK1 disrupts p53 signaling by increasing p53 protein levels without triggering downstream pro-apoptotic targets, enabling proliferation despite DNA damage [48]. This aberrant regulation allows DCLK1-positive cells to evade apoptosis and maintain survival, thereby fostering oncogenesis. Overexpression of DCLK1 elevates p53 protein while suppressing PUMA, BAX, and p21 [140]. DCLK1, or its effector p62, may interfere with p53 transcriptional activity or disrupt ATM/CHK signaling, promoting an apoptosis-resistant phenotype [141,142]. In tumors with p53 mutation or loss, DCLK1-positive CSCs bypass tumor suppression and initiate neoplastic transformation [143]. For example, DCLK1-positive tuft cells—typically tumor-suppressive—can become adenoma-initiating following APC loss or p53 dysfunction [144]. Thus, while DCLK1 supports epithelial regeneration by modulating p53, in malignancy it subverts this pathway to promote DNA repair, suppress apoptosis, and drive genomic instability.

### 3.5. GI Regeneration and Therapy Resistance Following IR

DCLK1-positive tuft cells are critical for recovery from IR-induced GI damage, driving crypt regeneration via enhanced DNA repair, anti-apoptotic signaling, and antioxidant defenses. These cells survive prolonged radiation exposure (>10 days post-irradiation), and their loss impairs crypt recovery, emphasizing DCLK1’s essential role in intestinal repair [145,146]. Radioprotection by DCLK1 involves apoptosis resistance mediated by NF-κB and Survivin, improving DNA repair through p53 and ATM [147]. DCLK1 orchestrates a network of stress response pathways. However, these survival mechanisms parallel those exploited by CSCs, contributing to tumor radioresistance and oncogenesis [36].

These CSCs rapidly repair DNA damage, maintain pro-survival signaling, evade apoptosis, and exhibit increased clonogenicity and stemness-associated gene expression. Silencing DCLK1 reduces these oncogenic attributes, highlighting its therapeutic relevance [49]. Chronic activation of DCLK1-driven pathways, including NF-κB, TGF-β, MAPK, and suppression of p53 checkpoints, promotes a regenerative yet inflammation-prone tumorigenic niche, facilitating transformation and recurrence post-radiation [30]. DCLK1 integrates cellular responses to oxidative stress and RAS signaling, mediated by Ang II and TGF-β, further connecting tissue repair with potential malignant fibrosis.

DCLK1-positive CSCs confer resistance to chemotherapy and RT via robust DNA repair, quiescence, apoptosis evasion, and modulation of the TME [148,149,150,151]. Elevated expression of DNA repair proteins and antioxidant enzymes enables efficient recovery from therapeutic damage [32]. Their quiescent state, maintained by cell cycle inhibitors, enables CSCs to evade therapies targeting rapidly proliferating cells and subsequently repopulate tumors upon reactivation [31,127,152]. Additionally, DCLK1-positive CSCs enhance survival by upregulating anti-apoptotic pathways and reducing pro-apoptotic signals [106,153]. Increased expression of ATP-binding cassette (ABC) transporters, notably P-glycoprotein (MDR1), further reduces chemotherapeutic effectiveness by actively effluxing drugs [154]. Clinically, radiation-induced GI toxicity varies by dose, fractionation, and anatomical location, requiring careful balancing of acute damage management with long-term cancer risk reduction [82]. DCLK1’s roles in epithelial regeneration and radioresistance underscore the necessity of targeted therapeutic approaches to optimize RT outcomes and limit ARS-related complications. Targeting DCLK1 thus emerges as a promising strategy to overcome CSC-driven resistance. Small-molecule inhibitors (e.g., DCLK1-IN-1) and monoclonal antibodies currently under preclinical evaluation demonstrate potential in sensitizing tumors to chemotherapy and RT without compromising normal tissue regeneration [35,106,113,152,155]. Future studies should validate these approaches in clinically relevant fractionated RT models and explore combination therapies with established radiosensitizers (PARP, ATR/ATM inhibitors, hypoxia modifiers) [49]. Overall, DCLK1 inhibition offers significant therapeutic promise for enhancing tumor radiosensitivity, reducing recurrence, and minimizing radiation-induced GI injury (Table 2).

## 4. IR-Induced Microenvironment and the Role of DCLK1

The GI tract is particularly susceptible to IR due to its rapid epithelial turnover and sensitive microenvironment. Radiation exposure disrupts the mucosal barrier, immune homeostasis, and gut microbiome [162], initially causing apoptosis and acute inflammation that impair epithelial regeneration [163]. Over time, chronic inflammation promotes fibrosis, vascular injury, and microbial dysbiosis [164,165,166,167,168]. Radiation-induced cell death releases danger-associated molecular patterns (DAMPs), which stimulate pro-inflammatory cytokines (IL-1β, TNF-α, IL-6) and recruit immune cells, such as neutrophils and macrophages, establishing a persistent inflammatory environment [169]. Sustained inflammation further encourages immunosuppression mediated by regulatory T cells and MDSCs, facilitating tissue remodeling and CSC emergence, thereby enhancing radioresistance [170,171].

DCLK1-positive tuft cells overlap functionally with CSCs, linking inflammation-induced tissue damage to CSC development. These cells exhibit superior DNA repair, drug efflux capabilities, and apoptosis evasion, thus contributing significantly to therapy resistance and tumor recurrence [32,172,173,174]. Clarifying these underlying mechanisms is critical for developing therapies that selectively protect healthy tissues while sensitizing tumors to radiation [175,176]. Notably, preclinical radiation models frequently utilize single high-dose exposures, differing from clinical radiotherapy protocols that employ fractionated doses administered over extended periods [177]. The regenerative response of DCLK1-positive tuft cells under fractionated radiation may differ substantially from their response to acute irradiation. Consequently, future preclinical studies must incorporate fractionated radiation models to accurately assess whether DCLK1 inhibition can sensitize tumors without compromising mucosal recovery in clinically relevant scenarios.

### 4.1. The Microenvironment in Gastrointestinal Acute Radiation Syndrome

A balanced interplay of pro- and anti-inflammatory responses is crucial for GI-injury progression. High-dose IR damages rapidly dividing immune cells—particularly lymphocytes—leading to immunosuppression and increased infection risk [178,179]. At the same time, IR activates macrophages and dendritic cells to release pro-inflammatory cytokines, exacerbating GI tissue damage [180,181]. Over time, regulatory T cells and MDSCs create an anti-inflammatory environment that suppresses immune surveillance, allowing mutated cells to escape detection and promoting angiogenesis, tissue remodeling, and cancer progression [182]. Macrophages play a central role in these processes by driving both inflammation and tissue repair, making them key therapeutic targets [183,184]. In the tumor microenvironment, macrophages often polarize toward an M2 phenotype that supports tumor survival, angiogenesis, and radioresistance while suppressing anti-tumor immunity [185]. Strategies that reprogram macrophages from an M2 to an M1 phenotype—such as inhibiting CSF-1 or targeting VEGF signaling—combined with immune checkpoint inhibitors, can reduce TAM-mediated radioresistance [37,186,187]. However, since macrophage-mediated repair is vital for normal tissue recovery, treatments must balance anti-tumor effects with the preservation of repair functions [188].

### 4.2. DCLK1 as a Modulator of the Radiation Microenvironment

DCLK1 plays a pivotal role in orchestrating the TME following IR injury, influencing immune cell polarization, cytokine secretion, extracellular matrix (ECM) remodeling, and redox balance. DCLK1-positive tuft cells, through the secretion of cytokines such as IL-25, activate group 2 innate lymphoid cells (ILC2s) to promote IL-4 and IL-13 release, which in turn drives macrophages toward an M2-like phenotype that favors tissue repair but also fosters immunosuppression and tumor progression [30,189]. In the irradiated microenvironment, this skewing of macrophages towards M2 polarization suppresses cytotoxic immune responses, enhances angiogenesis through vascular endothelial growth factor (VEGF) secretion, and promotes fibrosis via TGF-β signaling [190]. Furthermore, DCLK1 expression has been associated with the recruitment and activation of regulatory T cells (Tregs) and MDSCs, which together dampen anti-tumor immunity and facilitate tumor immune evasion [76].

Beyond its immunomodulatory functions, DCLK1 drives stromal remodeling; elevated serum DCLK1 correlates with increased TGF-β levels in fibrotic tissues, where TGF-β activates fibroblasts to deposit collagen and other ECM components, fostering hypoxia and a pro-tumorigenic niche [114]. In parallel, DCLK1 enhances CSC survival and resistance to oxidative stress [191]. Moreover, DCLK1 binds directly to IKKβ and induces its phosphorylation on Ser^177^/^181^ to activate NF-κB signaling, which drives chronic inflammation, upregulates DNA repair pathways, inhibits apoptosis, and promotes immunosuppression within the TME [105]. Together, these multifaceted actions establish DCLK1 as a master regulator of epithelial–stromal crosstalk. Therapeutic targeting of DCLK1 could sensitize tumors to radiotherapy while preserving normal mucosal repair, but strategies must precisely inhibit its pro-oncogenic and immunosuppressive signaling without abrogating essential regenerative functions [152]. Given these multifaceted roles, DCLK1 acts as a master regulator linking epithelial injury responses to microenvironmental remodeling, thereby balancing tissue regeneration with the risk of malignant transformation. Targeting DCLK1 in the context of radiation-induced damage holds promise not only for sensitizing tumors to radiotherapy but also for preserving normal tissue regenerative responses while mitigating the immunosuppressive and fibrotic consequences of IR exposure. However, therapeutic strategies must carefully balance DCLK1 inhibition to avoid impairing necessary mucosal repair processes while disrupting pro-oncogenic and immunosuppressive signaling.

### 4.3. Targeting DCLK1 to Overcome ARS-Induced Radioresistance and Secondary Cancer Risk

DCLK1 plays a key role in regulating TME and is a key factor in cancer stemness and radioresistance [30,192]. The TME comprises diverse cellular components and non-cellular elements that together coordinate CSC development [193]. It comprises diverse cellular and acellular components and coordinates CSC development. Myeloid cells—particularly monocytes and macrophages—support epithelial stemness while suppressing cytotoxic T-cell activity, thus promoting tumor progression and metastasis [194,195]. The M2 macrophage phenotype is closely linked to drug resistance, tumor progression, immunosuppression, EMT, and metastasis [196,197,198,199]. Immunosuppressive mechanisms, such as Treg and MDSC recruitment and PD-L1 expression, further dampen anti-tumor immunity, creating an inflammatory environment that drives CSC emergence and radioresistance [200,201,202]. Recent studies indicate that DCLK1 skews TAMs toward an M2-like state and promotes immunosuppressive cytokine secretion and Treg recruitment [189]. DCLK1-positive tuft cells also shape the tissue microenvironment by secreting cytokines like IL-25, which activates ILC2s to produce IL-13 [30]. This cascade polarizes macrophages toward the anti-inflammatory M2 phenotype, supporting tissue repair, extracellular matrix remodeling, and recovery from radiation-induced injuries [203,204]. In tumors, M2-polarized TAMs secrete VEGF-like factors to enhance blood supply and release IL-10 and TGF-β, which suppress cytotoxic T cells and expand Tregs [205]. Additionally, TAMs modulate redox balance by influencing antioxidant enzyme levels, thereby aiding DNA repair and angiogenesis [206]. DCLK1 further regulates NF-κB signaling to limit pro-inflammatory (M1) activation while promoting macrophage recruitment and activation in a manner that supports both tissue repair and tumor progression.

DCLK1-positive tuft cells are pivotal for GI regeneration after radiation injury [48]. In response to IR, tuft cell numbers increase (Figure 4), a phenomenon also observed in precancerous gastric lesions, although their numbers decrease in established gastric cancers [207,208]. Under chronic inflammatory and genotoxic stress, some tuft cells may transdifferentiate from a regenerative phenotype to a progenitor-like, neoplastic state [20]. This phenotypic plasticity, while critical for repair, may predispose surviving tuft cells to malignant transformation. Persistent DNA damage, aberrant signaling (e.g., via TGF-β and NF-κB), and an altered microenvironment collectively promote the emergence of tumor-initiating cells, thus increasing the risk of secondary cancers.

Preclinical GI-ARS models often use single high doses, whereas clinical radiotherapy delivers multiple fractions. Fractionated irradiation (e.g., five daily stereotactic body irradiation fractions) alters crypt survival, cytokine profiles, and regenerative kinetics; for instance, palbociclib had protective effects under single-dose IR but worsened injury under fractionation [61]. Thus, clinically relevant fractionated models are essential for evaluating DCLK1-mediated repair and immunomodulation. Preclinical models employing single high-dose irradiation were originally developed to simulate mass casualty scenarios involving acute, potentially lethal exposures to ionizing radiation—such as those that might occur following nuclear accidents or radiological terrorism. These models are intentionally designed to recapitulate the rapid onset and severity of GI-ARS in previously healthy subjects, rather than the prolonged and fractionated exposure patterns seen in clinical radiotherapy. As such, biological responses—including epithelial injury, crypt regeneration, and immune modulation—differ significantly between these two contexts. To better understand how fractionated radiation influences gastrointestinal tissue dynamics and the tumor microenvironment (TME), there is a critical need for updated preclinical models that incorporate clinically relevant dosing schedules. These models will be essential for evaluating how fractionation alters DCLK1-driven regenerative signaling, immune cell recruitment, and stromal remodeling, and for identifying therapeutic strategies that balance effective tumor control with mucosal preservation. Moreover, understanding how DCLK1 functions across both acute GI-ARS and fractionated RT injury models is vital for developing interventions that preserve mucosal regeneration while limiting radioresistance and secondary tumorigenesis.

## 5. GI-ARS Radioprotection, Mitigation, and Treatment

Radiation-induced GI injury represents a critical clinical challenge, characterized by epithelial stem cell depletion, barrier breakdown, and systemic inflammation [209]. Effective management requires a multifaceted approach incorporating radioprotectors, mitigators, and post-exposure treatments to preserve tissue integrity, support regeneration, and prevent long-term complications such as fibrosis and secondary malignancies [13,210,211]. Although several mitigators protect bone marrow, no FDA-approved agents currently reverse GI complications [212,213]. Once toxicity is established, interventions may alleviate symptoms, prevent sequelae, and promote repair. Below, we review current strategies for protecting the GI tract from radiation injury, highlight emerging therapeutic approaches, and discuss integrating DCLK1-targeted interventions within broader radioprotective frameworks.

### 5.1. Radioprotective and Radiomitigative Strategies

Radioprotectors—administered before or during irradiation—and radiomitigators—given shortly after exposure—aim to reduce oxidative stress, inflammation, and subsequent tissue damage. Classical agents such as Amifostine and Tempol scavenge ROS and enhance antioxidant defenses, while next-generation mimetics offer improved tissue specificity and reduced toxicity. Radiation induces pro-inflammatory cytokines (TNF-α, IL-1β, IL-6), promoting leukocyte infiltration and amplifying injury [214]. IR induces pro-inflammatory cytokines (TNF-α, IL-1β, IL-6), promoting leukocyte infiltration and amplifying injury [181]. Chronic inflammation activates TGF-β signaling, leading to excessive extracellular matrix deposition and fibrosis [215,216]. IR also stimulates the RAS, increasing angiotensin II levels, which, via AT_1_R, further intensify oxidative stress and fibrogenesis [217]. Clinically approved AT_1_R antagonists—losartan, telmisartan, and candesartan—reduce cytokine release and leukocyte infiltration [218]; in preclinical models, telmisartan mitigates fibrosis by decreasing collagen deposition and fibroblast activation [219,220]. Several antioxidant-based radiomitigators have shown efficacy in animal models. YK-4–250, a small molecule with antioxidant and anti-inflammatory properties, preserved GI barrier integrity and improved survival in irradiated mice [44]. Rx100, a lysophosphatidic acid analog that activates LPA_2_ receptors, inhibited apoptosis and enhanced survival in both mice and non-human primates [221]. While promising, these agents are not curative as monotherapies and likely require combination with regenerative or immunomodulatory interventions to achieve full therapeutic benefit.

When deployed as radiomitigators, ARBs synergize with antioxidants to both protect and restore tissues; combining telmisartan with Tempol reduces ROS, curbs fibrotic signaling, and supports epithelial regeneration [222]. However, some radioprotective agents can inadvertently enhance tumor cell survival [223]. Tumor cells, typically maintain high ROS levels due to elevated metabolism and oncogene activation, rely on ROS-dependent signaling for growth [224]. Exogenous antioxidants can disrupt these pathways by lowering ROS, increasing susceptibility to radiation-induced DNA damage, and furthering tumor progression [225,226]. Despite this risk, antioxidants benefit normal cells without compromising essential survival signals [227]. Moreover, tumor AT_1_R signaling promotes angiogenesis and metastasis; its blockade reduces tumor perfusion and dissemination, further sensitizing malignancies to IR [228,229]. Thus, integrating ROS scavengers with AT_1_R inhibition targets cancer-specific vulnerabilities while preserving normal tissues, enhancing the therapeutic index of radiotherapy without fostering radioresistance [230]. Additional investigational radiomitigators, including dipeptidyl peptidase-4 inhibitors, prostanoid analogs (e.g., misoprostol, 16,16-dimethyl PGE_2_), tocotrienols, nitroxide radicals, and γ-tocotrienol, demonstrate gut-protective effects in preclinical studies, with γ-tocotrienol advancing to non-human primate ARS testing. Superoxide dismutase mimetics also reduce acute and chronic GI toxicity in rodent and clinical radiotherapy settings [231].

### 5.2. Therapeutic Interventions for Established GI-ARS and Radiation-Induced GI Injury

GI-ARS arises from extensive crypt stem-cell loss, impaired mucosal regeneration, and barrier disruption, making it difficult to treat once established—unlike H-ARS, which often responds to intensive supportive care [232]. Treatment focuses on controlling inflammation, preserving barrier integrity, preventing infection, and promoting epithelial repair. Standard supportive measures—including IV fluids, electrolyte replacement, antiemetics, antidiarrheals, antibiotics, and parenteral nutrition—remain essential [233]. Overcoming the hematopoietic suppression requires growth factors; G-CSF and GM-CSF accelerate neutrophil recovery and reduce sepsis, while IL-11 alleviates thrombocytopenia and protects gut epithelium in preclinical models [234]. Adjunctive agents such as palifermin (keratinocyte growth factor), GLP-2 analogs, and glutamine have shown epithelial restorative potential in animal models, although their efficacy in fulminant GI-ARS remains under investigation [235,236,237].

#### 5.2.1. Senescence-Targeting Approaches

Radiation-induced senescence, driven by persistent DNA damage signaling via ATM and p53, leads to the upregulation of CDK inhibitors (p21^WAF1/CIP1^, p16^INK4A^) and a pro-inflammatory senescence-associated secretory phenotype (SASP). In the rapidly renewing intestinal epithelium, senescent cell accumulation impairs regeneration and perpetuates inflammation [238,239]. Senolytic agents such as Dasatinib + Quercetin and Navitoclax selectively eliminate p16^INK4A^ positive cells, promoting mucosal healing and reducing fibrosis and tumorigenesis in murine models without harming proliferative cells [240,241,242]. Senomorphics (e.g., JAK1/2 inhibitors) suppress SASP cytokines, restoring homeostasis [243]. When combined with radioprotectors (e.g., Tempol) or AT1 blockers (e.g., Telmisartan), senolytics may enhance tissue repair and reduce secondary cancer risk. By eliminating senescent “dead wood,” these therapies facilitate epithelial repopulation and reduce both acute and chronic injury.

#### 5.2.2. Anti-Fibrotic Strategies

Fibrosis often follows GI-ARS recovery, characterized by collagen deposition, strictures, and impaired absorption. Clinically, pentoxifylline + vitamin E and hyperbaric oxygen therapy (HBOT) modestly reverse fibrosis through TGF-β modulation and angiogenesis [244,245]. Preclinical agents—including statins (e.g., pravastatin), TGF-β-neutralizing antibodies, and pirfenidone—attenuate fibroblast activation and matrix deposition, warranting clinical trials [246,247,248]. TLR5 agonists such as entolimod and GI-targeted beclomethasone (OrbeShield^®^) improve survival and mucosal integrity in non-human primates when administered within 24 h post-irradiation [249,250]. Notably, entolimod reduced mortality 2–3-fold without intensive care and received an FDA Fast Track designation after successful Phase I trials [251,252]. These agents protect crypts by inducing endogenous prostaglandins and cytoprotective pathways.

#### 5.2.3. Stem Cell-Based and Regenerative Therapies

Mesenchymal stem cells (MSCs) are under investigation for radiation-induced proctitis. Early-phase trials and preclinical studies show that MSCs reduce inflammation, bleeding, and fibrosis by homing to injury sites and secreting trophic factors [253,254]. Wnt agonists and IL-22 analogs offer additional regenerative strategies. BCN057, a small-molecule Wnt/β-catenin activator, administered 24 h post-lethal abdominal irradiation, rescued Lgr5^+^ crypt stem cells and improved survival [232]. R-spondin1 similarly promoted crypt survival and regeneration [255]. IL-22-Fc, a dimerized analog, enhanced STAT3 activation in enterocytes and accelerated crypt regeneration after irradiation [256]. These agents selectively restore the intestinal stem cell niche without promoting tumor growth.

#### 5.2.4. DCLK1 and Radiation-Induced Senescence

Radiation-induced senescence is mediated by ATM-dependent activation of p53 and CDK inhibitors, enforcing permanent cell-cycle arrest and SASP. DCLK1^+^ tuft cells counteract this process by enhancing ATM phosphorylation and DNA repair, suppressing chronic senescence signaling [48]. Moreover, DCLK1 downregulates p21^WAF1/CIP1^ and p27^KIP1^, allowing cells to bypass arrest and maintain proliferative potential [257]. Through this dual mechanism—promoting repair and inhibiting senescence—DCLK1^+^ cells evade both apoptosis and senescence, contributing to regeneration but potentially increasing oncogenic risk.

## 6. Future Directions: Emerging Molecular Therapies

Radiation-induced GI damage remains a formidable clinical challenge due to its multifactorial pathogenesis—characterized by oxidative stress, inflammation, and impaired regeneration [63]. While current therapies aim to protect normal tissues, new strategies are needed to mitigate acute injury and long-term sequelae, including fibrosis and secondary malignancies. Emerging molecular interventions target key signaling pathways such as Nrf2, NF-κB, TGF-β, and p53 to preserve normal tissue integrity while limiting cancer stem cell (CSC) survival. Modulating Nrf2 with small molecules or gene therapy enhances antioxidant defenses and reduces ROS-induced damage [258]. NF-κB inhibitors may attenuate pro-survival and inflammatory signals, thereby limiting fibrosis without compromising immunity [259,260]. Novel TGF-β inhibitors preserve tissue architecture while minimizing fibrogenesis and tumor promotion [261]. Restoring p53 function enhances DNA repair and cell-cycle arrest, although caution is warranted to avoid supporting CSC viability [132]. Alternative strategies targeting mutant p53 offer a more tumor-selective approach [262,263]. These approaches—spanning small molecules, gene editing, and RNA interference—hold promise for improving the therapeutic index of radiotherapy.

### 6.1. Nanoparticle-Based Delivery Systems

Nanoparticle platforms enhance tissue specificity and reduce systemic toxicity of radioprotective agents via the enhanced permeability and retention (EPR) effect [264]. Ligand-modified polymeric nanoparticles can serve as a home to gastrointestinal or hematopoietic cells, increasing delivery precision [265]. Stimuli-responsive nanoparticles release cargo selectively in oxidative or acidic microenvironments typical of irradiated tissues [266], thereby concentrating therapeutic action while sparing healthy cells.

### 6.2. Modulating the Gut Microbiome and Innate Immunity

Emerging evidence underscores the critical role of the gut microbiome and innate immune responses in modulating radiation injury. Ionizing radiation induces dysbiosis and microbial translocation, exacerbating mucosal inflammation. Novel interventions now target microbial restoration and immune modulation as adjuncts to classical radioprotective strategies. One promising agent is MIIST305, a cationic glycopolymer that binds to the intestinal mucus layer and sequesters pro-inflammatory microbial products post-irradiation [267]. In murine models, oral administration of MIIST305 significantly improved survival following lethal radiation by reducing cytokine responses and promoting microbiome homeostasis. This highlights a broader immune-modulatory approach to mitigating GI damage induced by IR—one that extends beyond single cytokine inhibition to fortifying barrier integrity and restraining systemic inflammation.

### 6.3. Emerging Immunomodulatory Approaches

Immunomodulation represents a promising adjunct to radiotherapy. TAMs, often polarized to an M2 phenotype, promote angiogenesis and repair but suppress anti-tumor immunity and confer radioresistance [183,185]. Agents that reprogram TAMs toward an M1 phenotype—such as CSF-1R inhibitors or cytokine modulators—enhance phagocytic activity and pro-inflammatory cytokine production [186,187]. DCLK1^+^ tuft cells secrete IL-25, activating ILC2s and driving M2 polarization [189]. Targeting DCLK1 may shift the immune landscape toward a pro-inflammatory, tumor-suppressive phenotype. Combining radiotherapy with immune checkpoint inhibitors (e.g., anti-PD-1, anti-CTLA-4) further restores cytotoxic T-cell function and enhances tumor sensitivity to radiation.

### 6.4. Combined Drug Treatments

A multi-pronged approach is essential for mitigating both acute and delayed radiation damage. Radioprotectors (e.g., Tempol, MitoQ, SOD mimetics) reduce ROS-induced injury, while AT1 receptor blockers (e.g., Telmisartan, losartan) counteract RAS-mediated inflammation and fibrosis provides for one potential strategy. Hematopoietic growth factors (e.g., G-CSF) support marrow recovery (Table 3). DCLK1 inhibitors may reduce the risk of radiation-induced neoplasia by limiting the regeneration of damaged progenitor cells. Senolytic agents further suppress chronic inflammation and fibrosis. Integrating immunotherapies, such as TAM reprogramming and checkpoint blockade, may synergize with these strategies to restore immune surveillance and prevent tumor regrowth.

### 6.5. Integrating DCLK1 Inhibition with Fractionated Radiotherapy

Fractionated radiotherapy, grounded in the linear-quadratic model, remains a cornerstone of cancer therapy [268]. Evolving modalities—including hypofractionation, SBRT, and FLASH—continue to optimize therapeutic ratios [269]. Future studies should integrate DCLK1-targeted interventions into fractionation protocols. Preclinical models (e.g., patient-derived organoids, GEMMs) can assess DCLK1 inhibitors under varied dosing schemes to identify synergistic interactions. Clinically, early-phase trials might stratify patients by DCLK1 expression and monitor pharmacodynamic biomarkers to correlate pathway suppression with treatment response. Adaptive trial designs could define optimal dosing regimens that enhance radiosensitivity while preserving normal tissue repair.

### 6.6. Targeting DCLK1 to Enhance Radioprotection and Prevent Tumorigenesis

DCLK1 inhibition offers dual benefits: radiosensitizing tumors while protecting normal tissues. Preclinical studies show that DCLK1 blockade (via small molecules or siRNA) impairs DNA repair in CSCs, enabling tumor control at lower radiation doses [270]. This allows for dose de-escalation, minimizing normal tissue toxicity. DCLK1 inhibition may also suppress maladaptive regeneration, reducing the risk of radiation-induced tumorigenesis. CAR-T cells engineered with CBT-15 scFv represent a complementary strategy [27]. Radiation enhances DCLK1 surface expression and immune cell infiltration, creating a therapeutic window for CAR-T administration. Sequential delivery—radiotherapy, DCLK1 inhibition, followed by CBT-15 CAR-T infusion—could eradicate residual radioresistant CSCs. Preclinical optimization should explore variables including fraction size, inhibitor timing, CAR-T dosing, and lymphodepletion. Safety concerns (e.g., effects on normal tuft cells) may be addressed through suicide switches and restricted radiation fields. This integrated regimen may both enhance radioprotection and prevent long-term oncogenesis by targeting the DCLK1 axis.

### 6.7. DCLK1 as a Biomarker of Radiation Exposure and Regeneration 

DCLK1 expression in GI epithelia is dose-dependently induced by ionizing radiation, peaking 24–72 h post-exposure in murine models and correlating with crypt survival [271]. In colorectal cancer, high baseline DCLK1 expression predicts radioresistance and poor treatment response, supporting its use as a companion biomarker [272]. Circulating DCLK1^+^ extracellular vesicles mirror exposure severity and regenerative potential [273]. These vesicles may enable real-time, minimally invasive monitoring of GI damage progression and stratification for radioprotective interventions.

## 7. Conclusions and Perspectives

Radiation-induced GI damage presents a complex clinical challenge characterized by epithelial loss, barrier dysfunction, and sustained inflammation. Within this landscape, DCLK1 emerges as a pivotal regulator of both crypt regeneration and oncogenic potential. By promoting DNA repair, and pro-survival signaling, DCLK1 facilitates epithelial recovery, yet these same pathways may also confer stem-like traits that drive radioresistance, immune evasion, and tumorigenesis. Accordingly, DCLK1 represents both a therapeutic target and a biomarker of radiation injury. Quantifying tissue DCLK1 expression or tracking circulating DCLK1-positive extracellular vesicles could stratify injury severity, guide radioprotective interventions, and monitor therapeutic efficacy. Future studies should integrate DCLK1-targeted approaches with optimized fractionation, validated biomarker platforms, and combinatorial regimens involving radioprotectors, immunomodulators, and Senolytics to maximize therapeutic gain while minimizing long-term oncogenic risk.

Despite recent progress, critical gaps persist. Translational validation in large animal models and humans remains a top priority, as rodent models often fail to capture species-specific responses. Optimizing intervention windows and dosing regimens—particularly for agents with narrow therapeutic margins—will be essential for clinical implementation. Research must also delineate synergistic combinations that address the multifactorial pathology of IR induced GI damage, including oxidative stress, epithelial and vascular injury, dysbiosis, and immune dysregulation. Promising strategies include Wnt pathway agonists (e.g., R-spondin1) and growth factors (e.g., FGF-PT) that promote crypt regeneration, immune modulators like TLR5 agonists (e.g., entolimod) and cytokines (e.g., IL-22, IL-11) that enhance mucosal repair, and cell-based therapies such as MSCs and their exosomes, which confer regenerative and anti-inflammatory benefits. Molecularly targeted interventions offer complementary mechanisms by attenuating downstream injury pathways. Together, these approaches underscore the need for integrated, multimodal regimens tailored to injury stage and host status. Continued interdisciplinary research and regulatory engagement are essential to translating these insights into effective, scalable countermeasures for GI damage both intentionally and unintentionally induced.

## Figures and Tables

**Figure 1 cancers-17-02050-f001:**
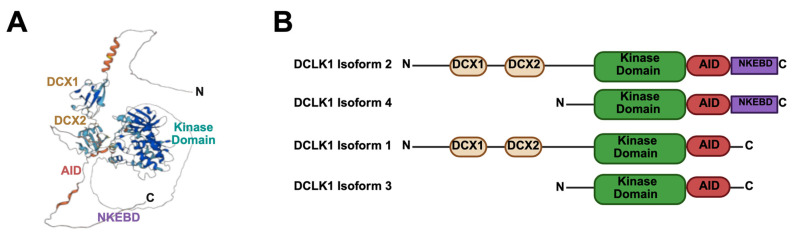
Structural organization and isoform diversity of human DCLK1. (**A**) Three-dimensional ribbon diagram of the DCLK1 protein illustrating key domains, including tandem doublecortin (DCX1, DCX2) microtubule-binding motifs, a serine/threonine kinase domain, an autoinhibitory domain (AID), and the C-terminal non-kinase extracellular binding domain (NKEBD) present in isoforms 2 and 4. Adapted from UniProt entry for DCLK1 (O15075) © UniProt Consortium. Accessed on 6 June 2025; https://www.uniprot.org/uniprotkb/O15075/entry. (**B**) Schematic representation of the four major human DCLK1 isoforms. Isoforms 1 and 2 contain two N-terminal DCX domains and a C-terminal kinase domain; isoforms 3 and 4 lack the DCX2 motif and initiate translation downstream. The autoinhibitory domain (AID) is conserved across all isoforms, while the NKEBD is restricted to isoforms 2 and 4. Isoform-specific expression governs subcellular localization and may determine differential roles in epithelial regeneration and tumorigenesis.

**Figure 2 cancers-17-02050-f002:**
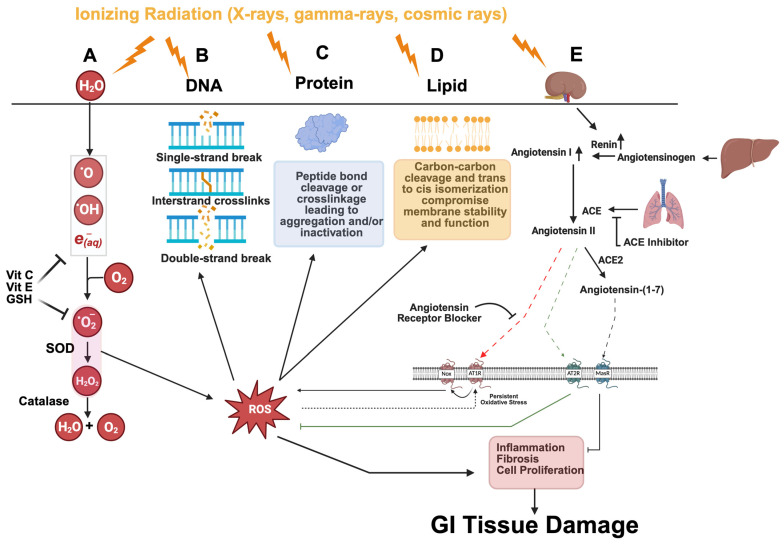
Ionizing radiation damage and ROS generation. (**A**) DCLK1^+^ cells enhance epithelial repair by activating antioxidant defenses, DNA damage responses, and regenerative cytokine signaling [31,32]. Ionizing radiation (IR) radiolysis water to generate ROS (O_2_•^−^, H_2_O_2_, •OH). Superoxide dismutase (SOD) converts O_2_•^−^ to H_2_O_2_, and catalase decomposes H_2_O_2_ into water and oxygen; nonenzymatic antioxidants (e.g., glutathione, vitamins C and E) also scavenge ROS. (**B**–**D**) IR and ROS induce DNA strand breaks and mutations, oxidize proteins (impairing function), and peroxidize lipids (compromising membrane integrity). (**E**) IR-induced oxidative stress upregulates renal renin, which converts hepatic angiotensinogen to angiotensin I. ACE (primarily in the lungs) then generates angiotensin II (Ang II). Ang II activates NADPH oxidases via AT1R, producing additional ROS and triggering inflammatory, fibrotic, and proliferative pathways that worsen tissue injury; feedback loops sustain ROS production and inflammation. By contrast, Ang II binding to AT2R attenuates inflammation, and ACE2 converts Ang II to angiotensin-(1–7), which signals anti-inflammatory effects via MasR. Clinically, ACE inhibitors block Ang II formation, and ARBs prevent its binding to AT1R (Created in BioRender. Moore, L. (2025) https://app.biorender.com/illustrations/67bf8f06194e14d85df5a26d?slideId=acc0c132-3b80-4da9-bd17-c95bcfd7e2fc.

**Figure 3 cancers-17-02050-f003:**
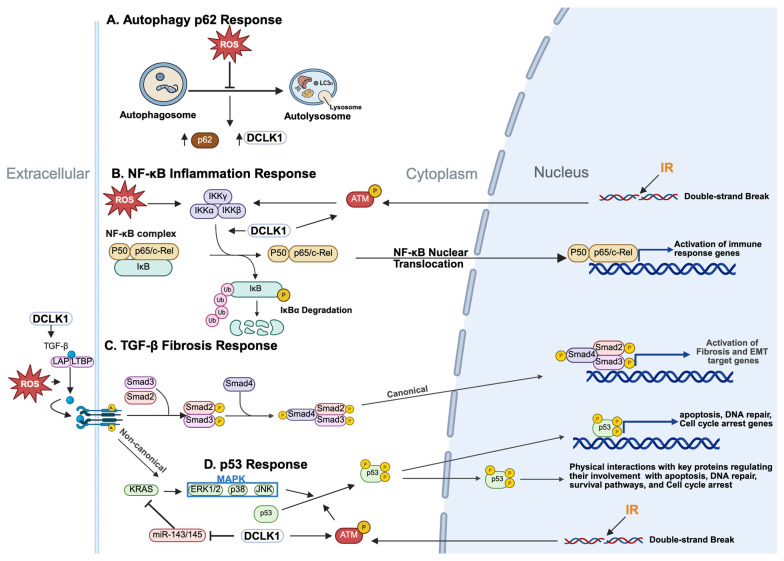
DCLK1 cellular and molecular mechanisms underlying IR-mediated radioresistance. The key processes by which cells enhance resistance to radiation-induced damage. IR initiates the sequence by inducing ROS through interactions with water molecules and damage to cellular components. (**A**) Autophagy–p62 Axis: IR-induced ROS disrupt autophagic flux, causing p62/SQSTM1 accumulation and reduced degradation of DCLK1. Concurrent stabilization of DCLK1 further activates stress-response pathways that enhance cell viability and redox balance but may also contribute to oncogenic signaling. (**B**) NF-κB Inflammatory Response: DCLK1 promotes the phosphorylation of IKKβ, leading to the degradation of IκB and nuclear translocation of NF-κB. NF-κB activation upregulates genes involved in inflammation, DNA repair, and cell survival, supporting adaptation to radiation-induced damage. (**C**) TGF-β Fibrosis and EMT Response: DCLK1 enhances TGF-β signaling triggered by ROS. In the canonical pathway, TGF-β activates SMAD-dependent transcription of genes involved in fibrosis and epithelial–mesenchymal transition (EMT). DCLK1 promotes EMT by upregulating transcription factors such as ZEB1/2 and SNAIL, contributing to tissue remodeling, fibrogenesis, and resistance to therapy. (**D**) p53 Response: DCLK1 phosphorylates ATM leading to p53 phosphorylation. p53~P through either the TGF-β non-canonical pathway or in response to DNA damage resulting from IR is hyperphosphorylated to promote the activation of proteins involved in DNA repair, cell cycle arrest, and apoptosis. Additionally, DCLK1 suppresses miR-143/145, a tumor-suppressive miRNA cluster that normally inhibits KRAS. Loss of miR-143/145 derepresses KRAS and activates downstream MAPK signaling (ERK, JNK, p38), further supporting proliferation, stress adaptation, and radiation resilience (Created in BioRender. Moore, L. (2025) https://app.biorender.com/illustrations/684b689e0656afea9db5995f?slideId=ae9d746e-eb88-4727-bbc7-f1124f8343e0).

**Figure 4 cancers-17-02050-f004:**
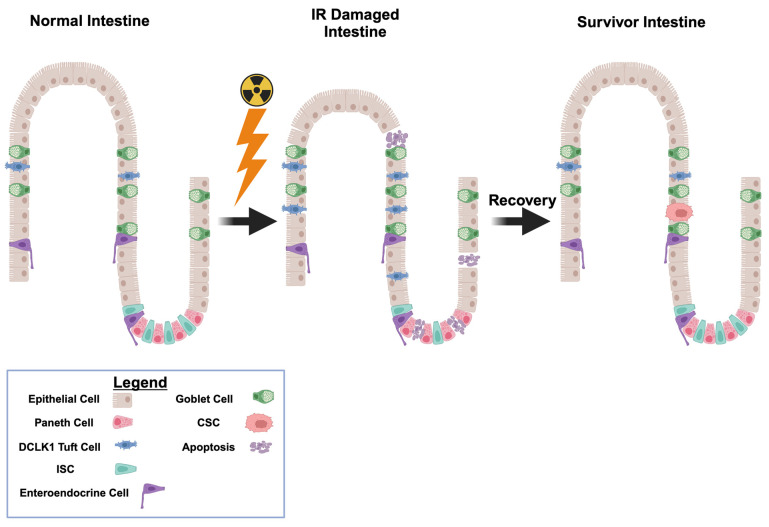
DCLK1 and the dual role of tuft cells in GI regeneration and tumor radioresistance. The chronic inflammatory milieu following ionizing radiation exposure fosters an environment conducive to cancer stem-like cell (CSC) development. The lightning bolt icon denotes uniform IR exposure to all cells. Radiation-induced damage triggers a sustained inflammatory response that elevates DCLK1-positive tuft cell levels to support tissue regeneration. Fragmented cells represent apoptotic figures that will be replaced during recovery. DCLK1^+^ tuft cells, which rapidly respond to IR by (1) activating antioxidant defenses and (2) initiating ATM-mediated p53 stabilization to drive DNA repair. These protective signals can rescue damaged cells. Tuft cells also secrete regenerative cytokines and proliferate, enabling the replacement of any cells lost to apoptosis. Together, these DCLK1-driven pathways limit ROS-mediated injury, restore genomic integrity, and repopulate the epithelium, thereby conferring radioresistance. However, under persistent inflammatory and genotoxic stress, these same tuft cells may undergo transdifferentiation, acquiring CSC properties characterized by enhanced DNA repair, drug efflux, and radioresistance. Consequently, while DCLK1-positive cells are critical for normal GI repair, their dual role can also contribute to secondary malignancies in radiation survivors (Created in BioRender. Moore, L. (2025) https://app.biorender.com/illustrations/676ed4518e51ca549e200ff6?slideId=30fa7e8b-ee18-4d16-bd21-97272cf19a43).

**Table 1 cancers-17-02050-t001:** Organ-specific ARS effects based on radiation dose.

Dose Range (Gy)	Affected Syndrome	Organ-Specific Damage	Associated Symptoms	Overlapping Effects	Studies Summary
1–2 Gy	Hematopoietic (H-ARS)	Mild bone marrow suppression, reduced blood cell counts	Fatigue, mild nausea, increased infection risk	Latent period may mask hematologic effects	Preclinical [13]
2–7 Gy	Hematopoietic (H-ARS)	Severe bone marrow suppression, pancytopenia	Hemorrhage, infections, anemia, fever	GI symptoms (nausea, vomiting) may appear at higher end	Preclinical [13] Clinical [9]
7–15 Gy	GI-ARS, H-ARS	Destruction of GI epithelial lining, crypt loss, mucosal barrier breakdown	Severe diarrhea, dehydration, abdominal pain, septicemia	Hematopoietic damage worsens GI injury, increasing lethality	Preclinical [13] Clinical [9]
15–30 Gy	GI-ARS, CNS effects	Massive GI damage, early CNS dysfunction	Severe GI symptoms, neurovascular instability	Hematopoietic failure exacerbates multi-organ dysfunction	* Preclinical [13] Clinical [6]
>30 Gy	CNS-ARS, GI-ARS	Brain edema, neuronal death, blood–brain barrier disruption, irreversible GI damage	Headache, confusion, seizures, loss of consciousness, death within hours to days	GI and hematopoietic syndromes accelerate neurovascular instability	Clinical [6]

Table 1 compiles dose-dependent effects from both preclinical (mouse, rat) and human studies. *—In rodents, GI-ARS is typically studied up to 10–12 Gy—higher regional or fractionated doses (e.g., 15 Gy to the abdomen) cause massive mucosal necrosis and early neurologic signs (ataxia, lethargy), and single-fraction whole-body exposures above ~20 Gy result in rapid CNS collapse and irreversible GI failure.

**Table 2 cancers-17-02050-t002:** Protective vs. oncogenic signaling influenced by DCLK1.

Pathway	Protective Effect	Oncogenic Effect
NF-κB	Transient NF-κB activation promotes pro-survival cytokines and regeneration [156]	Sustained NF-κB signaling drives inflammation, immune evasion, and tumor progression [157]
TGF-β	TGF-β–mediated EMT in repair phase facilitates wound closure and restitution [158]	Persistent TGF-β signaling leads to EMT-driven invasion, fibrosis, and metastatic potential [159]
p53	Modulates DNA repair and cell-cycle checkpoints preserves genomic integrity in surviving cells [160]	Inhibition of p53–mediated apoptosis allows survival of cells with oncogenic lesions [143]
MAPK	Activation of ERK/JNK pathways drives proliferation of crypt progenitors and regeneration [161]	Chronic MAPK signaling leads to uncontrolled proliferation, survival, and resistance to apoptosis [118]

**Table 3 cancers-17-02050-t003:** Combination strategy to combat ARS.

Combination Strategy	Drug Combination/Example	Mechanism/Rationale
Antioxidant + AT1 Receptor Blocker	Tempol (or MitoQ/SOD mimetics) + Telmisartan (or Losartan/Candesartan)	Antioxidants reduce ROS-induced damage while ARBs block angiotensin II–mediated inflammation and fibrosis, protecting both GI and hematopoietic tissues.
DCLK1 Inhibition + Antioxidant	DCLK1-targeting agent (experimental) + Tempol or MitoQ	Inhibiting DCLK1 prevents protection of cancer stem-like cells (CSCs) and, when combined with antioxidants, helps preserve normal tissue regeneration while reducing therapy resistance and tumor recurrence.
Senolytic Therapy + Radioprotective Agents	Dasatinib + Quercetin (or Fisetin/Navitoclax) combined with Tempol and/or Telmisartan	Senolytics clear radiation-induced senescent cells to reduce chronic inflammation and fibrosis; the addition of antioxidants/ARBs improves tissue repair and mitigates long-term complications.
Immunomodulatory Combination	CSF-1R inhibitor + Immune checkpoint inhibitors (e.g., anti-PD-1) + ARB	This approach reprograms tumor-associated macrophages (TAMs) from an M2 to M1 phenotype and restores anti-tumor immunity while protecting normal tissue repair mechanisms.
Supportive Hematopoietic Growth Factor Integration	Granulocyte Colony-Stimulating Factor (G-CSF) added to the above regimens	G-CSF aids in restoring bone marrow function and immune competence, which is critical given the interconnected hematopoietic and GI injuries observed in ARS.

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
