# Peer review of "Balancing Regeneration and Resistance: Targeting DCLK1 to Mitigate Gastrointestinal Radiation Injury and Oncogenesis"

_cancers, 2025, doi:10.3390/cancers17122050_

Round 1

Reviewer 1 Report

Comments and Suggestions for Authors

This is a well written and comprehensive review of the role of DCLK1+ tuff cells in regeneration and potential cancer stem cell induction in GI epithelium in response to ionizing radiation (IR) injury. The authors have also incorporated an excellent review of acute radiation syndrome (ARS) and its sub-syndromes including GI-ARS. The discussion also includes a very complete review of radioprotectants and mitigators in this space.

This is an informative review and enjoyable to read; however, I do note the following comments/concerns that should be addressed:

  1. Regarding lines 360-366.

The intent of preclinical models employing single large acute doses of IR is to simulate lethal, or potentially lethal, IR exposures that might be received in mass casualty incidents rather than exposures received therapeutically. The two types of exposure and their consequences are well established as not being equivalent. It is not clear that the preclinical models referenced here should be considered relevant to clinical RT at all. In fact, in models of GI-ARS, GI-ARS is typically considered to arise in response to acute high dose IR exposures in the GI tissue of subjects that were otherwise healthy prior to IR exposure.

As such, the reasons the preclinical models noted here were developed, and the way they are employed should be explained here as a justification for why new preclinical irradiation models should be developed to investigate fractionated RT effects on the GI in the context of cancer RT. The tissue repair and recovery that fractionation facilitates may be an important factor in microenvironment and the TME immunological responses discussed in the following sections of this review.

  1. Regarding lines 455-458.

The temporally protracted dose delivery of fractionated RT does not fit the definition of an acute exposure; thus, this use of the term GI-ARS is incorrect. Throughout the manuscript the authors conflate acute high dose IR exposure with fractionated high dose RT. As the authors themselves point out, these two scenarios are unlikely to result in biologically equivalent outcomes. Some separation in terminology needs to be incorporated into the text to differentiate the two conditions.

Minor edits:

  1. All 3 tables are poorly formatted and difficult to read.
  2. Figure 2 legend, lines 215-216: This sentence is poorly constructed and doesn't make sense as written.
  3. Figure 2 legend, lines 218-221: The lower case "b" used here with TGF-b should be replaced with the Greek letter beta for accuracy and consistency with the manuscript text.
  4. Line 395: The abbreviation MDSC should be used here, as MDSC has previously been defined.
  5. Line 1187 (citation line 497): Reference 212 and 33 are duplicates.
  6. Line 1326 (citation line 665): Reference 269 and 106 are duplicates.
  7. Line 1315 (citation line 647): Reference 264 is incomplete.
  8. Please review the references for additional duplicates, as my review may have missed others.

Author Response

  1. The intent of preclinical models employing single large acute doses of IR is to simulate lethal, or potentially lethal, IR exposures that might be received in mass casualty incidents rather than exposures received therapeutically. The two types of exposure and their consequences are well established as not being equivalent. It is not clear that the preclinical models referenced here should be considered relevant to clinical RT at all. In fact, in models of GI-ARS, GI-ARS is typically considered to arise in response to acute high dose IR exposures in the GI tissue of subjects that were otherwise healthy prior to IR exposure. As such, the reasons the preclinical models noted here were developed, and the way they are employed should be explained here as a justification for why new preclinical irradiation models should be developed to investigate fractionated RT effects on the GI in the context of cancer RT. The tissue repair and recovery that fractionation facilitates may be an important factor in microenvironment and the TME immunological responses discussed in the following sections of this review. Response: We revised to, “Preclinical GI-ARS models often use single high doses, whereas clinical radiotherapy delivers multiple fractions. Fractionated irradiation (e.g., five daily stereotactic body irradiation fractions) alters crypt survival, cytokine profiles, and regenerative kinetics; for instance, palbociclib protected under single-dose IR but worsened injury under fractionation [1]. Thus, clinically relevant fractionated models are essential for evaluating DCLK1-mediated repair and immunomodulation. Preclinical models employing single high-dose irradiation were originally developed to simulate mass casualty scenarios involving acute, potentially lethal exposures to ionizing radiation—such as those that might occur following nuclear accidents or radiological terrorism. These models are intentionally designed to recapitulate the rapid onset and severity of GI-ARS in previously healthy subjects, rather than the prolonged and fractionated exposure patterns seen in clinical radiotherapy. As such, biological responses—including epithelial injury, crypt regeneration, and immune modulation—differ significantly between these two contexts. To better understand how fractionated radiation influences gastrointestinal tissue dynamics and the tumor microenvironment (TME), there is a critical need for updated preclinical models that incorporate clinically relevant dosing schedules. These models will be essential for evaluating how fractionation alters DCLK1-driven regenerative signaling, immune cell recruitment, and stromal remodeling, and for identifying therapeutic strategies that balance effective tumor control with mucosal preservation. Moreover, understanding how DCLK1 functions across both acute GI-ARS and fractionated RT injury models is vital for developing interventions that preserve mucosal regeneration while limiting radioresistance and secondary tumorigenesis.
  2. The temporally protracted dose delivery of fractionated RT does not fit the definition of an acute exposure; thus, this use of the term GI-ARS is incorrect. Throughout the manuscript the authors conflate acute high dose IR exposure with fractionated high dose RT. As the authors themselves point out, these two scenarios are unlikely to result in biologically equivalent outcomes. Some separation in terminology needs to be incorporated into the text to differentiate the two conditions. Response: We have revised the document in numerous places to clarify when it is referring to GI-ARS or clinical treatment, specifically we document whether this issue is acute response to GI-ARS or if it is a response to IR-induced GI injury, that may be acute or chronic.

Minor edits:

  1. All 3 tables are poorly formatted and difficult to read. Response: We apologize for not checking the formatting.
  2. Figure 2 legend (now Figure 3 Legend), lines 215-216: This sentence is poorly constructed and doesn't make sense as written. Response: Due to suggested revisions to the figure, we composed a new legend.
  3. Figure 2 legend, lines 218-221: The lower case "b" used here with TGF-b should be replaced with the Greek letter beta for accuracy and consistency with the manuscript text. Response: We changed to Greek character.
  4. Line 395: The abbreviation MDSC should be used here, as MDSC has previously been defined. Response: We edited to the abbreviation as recommended.
  5. Line 1187 (citation line 497): Reference 212 and 33 are duplicates. Response: We removed the redundant reference.
  6. Line 1326 (citation line 665): Reference 269 and 106 are duplicates. Response: We removed ref 269 as it is not needed.
  7. Line 1315 (citation line 647): Reference 264 is incomplete. Response: We updated this book chapter reference.
  8. Please review the references for additional duplicates, as my review may have missed others. Response: We have reviewed and removed duplicate references.

We thank the reviewer for their thoughtful and constructive feedback, which significantly improved the clarity and accuracy of the manuscript. In particular, the reviewer’s comments helped refine our discussion of radiation models and prompted important distinctions in terminology surrounding GI-ARS and fractionated GI injury. We also appreciate the attention to detail in identifying formatting inconsistencies and reference redundancies, which have now been corrected. This feedback was instrumental in enhancing both the scientific rigor and presentation quality of the revised manuscript.

Reviewer 2 Report

Comments and Suggestions for Authors

This is a review paper that discusses some of the current problems with radiation therapy.  While radiation therapy provides benefit to patients with malignancies, it can also cause substantial collateral damage and second tumors.

The focus of the article is on the role of the DCLK1 pathway in these processes

The article is well written and provides an in-depth review.

If there is any way to shorten the manuscript, it would improve the readability.

Otherwise, it is acceptable. 

Author Response

This is a review paper that discusses some of the current problems with radiation therapy.  While radiation therapy provides benefit to patients with malignancies, it can also cause substantial collateral damage and second tumors. The focus of the article is on the role of the DCLK1 pathway in these processes The article is well written and provides an in-depth review. If there is any way to shorten the manuscript, it would improve the readability. Otherwise, it is acceptable. Response: We sincerely thank the reviewer for their positive assessment of our manuscript and their recognition of its clarity and depth. We appreciate the constructive suggestion to improve readability by streamlining the text. In response, we carefully reviewed the manuscript for opportunities to condense or refocus content—particularly in sections covering broader therapeutic strategies—to better emphasize DCLK1-specific mechanisms. These refinements were made with the goal of enhancing clarity while preserving the mechanistic detail and translational relevance. We are grateful for the reviewer’s thoughtful input and support.

Reviewer 3 Report

Comments and Suggestions for Authors

Ionizing radiation (IR) remains a cornerstone of cancer therapy but poses risk to normal tissues, causing acute radiation syndrome (ARS). Doublecortin-like kinase 1 (DCLK1)-positive tuft cells promote mucosal regeneration via antioxidant defenses, DNA repair, and pro-survival signaling; however, these features also confer cancer stem cell–like properties, driving radioresistance, immune evasion, and tumorigenesis. The authors reviewed the involvement of DCLK1 in Nrf2-mediated antioxidant responses, NF-κB-driven inflammation, TGF-β-induced epithelial–mesenchymal transition, MAPK cascades, and p53 pathways, linking repair with oncogenic risk, and modulates the tumor microenvironment. There are issues to consider:

  1. Line 135. “the renin-angiotensin system (RAS)” should show the abbreviation only.
  2. Line 216, 217. “NF-kb” should be “NF-κB”.
  3. Line 218, 221. TGF-b” should be “TGF-β”.
  4. The mechanisms of DCLK1 regulated Nrf2-mediated antioxidant responses under Ionizing radiation in cancer cells are unclear in the manuscript. please illustrate the mechanisms of DCLK1-mediated Nrf2 or NF-κB or TGF-β signaling underlying ARS-mediated radioresistance.
  5. Line 400. The authors claimed that “in parallel, DCLK1 sustains antioxidant defenses by sequestering p62 and promoting Nrf2/KEAP1–mediated transcription of cytoprotective genes, thereby enhancing CSC survival and resistance to oxidative stress [181]”. Although the author also cited the reference, “Co-localization of autophagy-related protein p62 with cancer stem cell marker dclk1 may hamper dclk1's elimination during colon cancer development and progression”, there is still lack of the evidence that DCLJ1 regulates Nrf2-mediated radioresistance in tumor microenvironment. Please cite the original article.
  6. Line 303. The authors claimed that “Radioprotection by DCLK1 involves apoptosis resistance mediated by NF-κB and Survivin, improved DNA repair through p53 and ATM, and antioxidant activation via Nrf2 [140].” However, NF-κB and Survivin were not mentioned in the reference #140. Please cite the original article.

Author Response

  1. Line 135. “the renin-angiotensin system (RAS)” should show the abbreviation only. Response: We have updated Line 135 to read simply “RAS” (removing the parenthetical abbreviation) and ensured that “renin‐angiotensin system” appears in full only at its first mention.
  2. Line 216, 217. “NF-kb” should be “NF-κB”. Response: We apologize for the oversight and have replaced both occurrences of “NF-kb” with “NF-κB” (Lines 216–217).
  3. Line 218, 221. TGF-b” should be “TGF-β”. Response: Both instances have been corrected to “TGF-β” (Lines 218 and 221).
  4. The mechanisms of DCLK1 regulated Nrf2-mediated antioxidant responses under Ionizing radiation in cancer cells are unclear in the manuscript. please illustrate the mechanisms of DCLK1-mediated Nrf2 or NF-κB or TGF-β signaling underlying ARS-mediated radioresistance. Response: We have updated the figure (now figure 3) and expanded the text to clarify DCLK1’s roles under IR. First, we show how IR-induced ROS impairs autophagy, drives p62–DCLK1 co-localization, and stabilizes Nrf2 to boost antioxidant defenses. Second, we demonstrate that DCLK1 activates IKKβ, leading to IκB phosphorylation and NF-κB (p65/p50)–mediated transcription of anti-apoptotic genes. Finally, we note that DCLK1 expression correlates with TGF-β levels and that DCLK1 directly binds ATM following IR, linking it to TGF-β–driven responses and DNA-damage repair.
  5. Line 400. The authors claimed that “in parallel, DCLK1 sustains antioxidant defenses by sequestering p62 and promoting Nrf2/KEAP1–mediated transcription of cytoprotective genes, thereby enhancing CSC survival and resistance to oxidative stress [181]”. Although the author also cited the reference, “Co-localization of autophagy-related protein p62 with cancer stem cell marker dclk1 may hamper dclk1’s elimination during colon cancer development and progression”, there is still lack of the evidence that DCLJ1 regulates Nrf2-mediated radioresistance in tumor microenvironment. Please cite the original article. Response: In response to the reviewer’s point, we acknowledge that no study to date directly demonstrates DCLK1 regulating Nrf2‐mediated radioresistance in the tumor microenvironment. Roy et al. demonstrated that, when autophagy is pharmacologically blocked, p62 and DCLK1 co‐localize, impairing DCLK1 turnover and sustaining Keap1 sequestration, which in turn would stabilize Nrf2 and increases antioxidant gene expression. To our knowledge, no published work yet shows this mechanism operating directly under IR exposure within the tumor microenvironment. To avoid overstating, we have revised the text and legend to note that, while indirect evidence of a DCLK1–p62–Keap1–Nrf2 axis exists, formal proof of DCLK1 driving Nrf2‐mediated radioresistance in vivo is still lacking. Specifically, we now write: “However, no study to date directly demonstrates DCLK1 regulating Nrf2‐dependent radioresistance within the tumor microenvironment.”
  6. Line 303. The authors claimed that “Radioprotection by DCLK1 involves apoptosis resistance mediated by NF-κB and Survivin, improved DNA repair through p53 and ATM, and antioxidant activation via Nrf2 [140].” However, NF-κB and Survivin were not mentioned in the reference #140. Please cite the original article. Response:

We believe these revisions fully address the reviewer’s concerns by (a) clarifying the molecular mechanisms by which DCLK1 modulates Nrf2, NF-κB, and TGF-β in the context of IR, and (b) ensuring that all pathway claims are supported by original, peer‐reviewed evidence. Thank you for helping us improve the accuracy and rigor of our manuscript.

Reviewer 4 Report

Comments and Suggestions for Authors

Comments to authors:

The manuscript of an article, which was written by Dr Landon L Moore et al, is interesting, discussing if Coublecortin-like kinase 1 (DCLK1) could be a potential therapeutic target for GI-ARS. However, I suggest authors edit text and Figures. Most important is that authors did not describe the background for DCLK1 in the Introduction section. Readers, except those who are studying DCLK1, will wonder about the molecular structure and biological functions of the protein. How was the protein or the encoding gene identified? What factors are phosphorylated by the DCLK1? These kinds of backgrounds with some simple illustrations would be explained in the first place. I, therefore, would suggest the authors edit the text and add Figures. Moreover, I would suggest authors make the best effort to eliminate descriptions that are not associated with DCLK1.

Recommendation: Major revision

General comments

Although authors described that DCLK1 plays a role in post-IR repair as a central mediator, its molecular structure and biological function are not explained. I advise authors to add a picture showing the molecular structure and the essential functions of DCLK1. That will greatly help readers comprehension why the protein plays a role in the response to IR stress. Moreover, before discussing the biological roles of DCLK1 in GI-ARS, description about the process of how the protein or the encoding gene was identified could be added. Additionally, the related proteins/genes and DCLK1-interactiong factors can be provided in the first place.

Specific comments

  1. P3: Table 1 might better be edited providing appropriate citations. It would be indicated if the effects of each dose of the Radiation were obtained from animal experiments or clinical databases. In addition, the Radiation dose that can affect DCLK1 would be shown.
  2. P4, Figure 1: Please indicate when and where the DCLK1 functions to affect responses during the GI Tissue Damage-causing process.
  3. P6, L207-210: Appropriate references should be indicated.
  4. P6, Figure 2: In this Figure, DCLK1 is not indicated. I suggest authors put DCLK1 with the interacting proteins on the suitable part.
  5. P6 and 7, Figure 2 legend: Nf-kb and TGF-b; NF-kB and TGF-b.
  6. P7, L233-235 and L262-263, and P8, L270-272: Regarding the reference [99] and others, if DCLK1 mediated signaling system were shown experimentally or even if it were just speculation, that can be illustrated.
  7. P8, L273-297: It is interesting that DCLK1 functions as a epithelial generation factor with p53-modulation functions. It would be worth depicting the molecular mechanisms.
  8. P9, Table 2: Appropriate references should be indicated.
  9. P10, L367-383: This section seems to be not directly associated with DCLK1. Therefore, I recommend authors to edit.
  10. P12, Figure 3: The legend is alright. However, this Figure is hard to understand how DCLK1 plays a role in Radioresistance. Indicate IR-damage not on the arrow but on each cell. Additionally, how do apoptotic cells recover during the process? That should be explained in the legend.
  11. P13, L481-P14, L577: The description is too long. The only part that contains description on DCLK1 is 5.2.4. Authors had better make best effort to edit the section as concise as possible.
  12. P16, Table 3: Appropriate references should be indicated. Some of the chemical compounds or drugs, including TEMPOL, should be explained for readers to understand why they were used for the combinatorial treatment for ARS.

Author Response

  1. … authors did not describe the background for DCLK1 in the Introduction section. Readers, except those who are studying DCLK1, will wonder about the molecular structure and biological functions of the protein. How was the protein or the encoding gene identified? What factors are phosphorylated by the DCLK1? These kinds of backgrounds with some simple illustrations would be explained in the first place. therefore, would suggest the authors edit the text and add Figures. Response: We fully agree and thank the reviewer, as such we added back more background on DCLK1, including adding a figure. Specifically in the Introduction, a paragraph outlining the identification and cloning of DCLK1, its structure and isoforms, as well as the limited understanding of its direct substrates, but the broad involvement in key pathways. To further aid comprehension, we have added a new Figure 1 that illustrates the domain structure of DCLK1 and distinguishes its major isoforms (1–4), including the autoinhibitory domain (AID) and the non-kinase extracellular binding domain (NKEBD) specific to isoforms 2 and 4. This schematic highlight structural features relevant to DCLK1’s function in tissue regeneration and its therapeutic potential. These additions provide a foundational framework for understanding DCLK1’s role in the context of radiation-induced GI injury and align with the reviewer’s recommendation to present this information “in the first place.”
  2. I would suggest authors make the best effort to eliminate descriptions that are not associated with DCLK1. Response: We appreciate the reviewer’s suggestion to streamline the manuscript by focusing on DCLK1-specific content. However, we respectfully maintain that the inclusion of broader contextual information—such as epithelial injury mechanisms, immune modulation, and fibrosis—is essential to understanding the physiological landscape in which DCLK1 operates. These descriptions provide the mechanistic framework that enables readers to grasp how DCLK1 integrates into complex post-radiation tissue responses. For clarity and focus, we have revised several sections to emphasize DCLK1’s role more explicitly and have trimmed general therapeutic content where appropriate. Nevertheless, we believe that retaining key background on radiation-induced GI injury and associated signaling pathways is necessary to support the review’s central thesis: that DCLK1 serves as a mechanistically distinct and therapeutically actionable node in radiation response.
  3. P3: Table 1 might better be edited providing appropriate citations. It would be indicated if the effects of each dose of the Radiation were obtained from animal experiments or clinical databases. In addition, the Radiation dose that can affect DCLK1 would be shown. Response: We thank the reviewer for this suggestion. Table 1 now indicates whether findings are from preclinical or clinical studies, prioritizing review articles to minimize citations and specifying animal versus human data to enhance transparency without compromising the table’s comparative value. Additionally, we concluded this section with the following, “Recent work shows that DCLK1‐expressing tuft cells play a critical role in post‐irradiation repair: in mice, a single 12 Gy total‐body dose leads to robust Dclk1 upregulation in surviving crypts, activation of DNA‐damage and antioxidant pathways, and improved epithelial regeneration—effects that mitigate GI-ARS severity [2]. Similarly, in vitro experiments using colorectal cancer lines reveal that silencing DCLK1 increases radiosensitivity at doses around 6 Gy, underscoring DCLK1’s function in DNA repair and cell-survival signaling after moderate IR exposure [3].” To better integrate DCLK1 in ARS research.
  4. P4, Figure 1: Please indicate when and where the DCLK1 functions to affect responses during the GI Tissue Damage-causing process. Response: We thank the reviewer for this suggestion. In the revised manuscript, the referenced schematic is now Figure 2, which illustrates gastrointestinal injury and regeneration after ionizing radiation. Although DCLK1 expression increases during the early regenerative phase—particularly in tuft cells—its precise mechanisms remain undefined. DCLK1⁺ cells support epithelial repair via antioxidant defenses, DNA damage responses, and regenerative cytokine signaling [2,3]. Because the schematic already captures this dynamic and incorporating undefined molecular details would overcomplicate the figure, we did not alter its visuals. Instead, we clarified the timing of DCLK1 activation in the Figure 2 legend to emphasize its role during post-injury regeneration.
  5. P6, L207-210: Appropriate references should be indicated. Response: We appreciate the reviewer’s feedback and have revised the relevant section to clarify and strengthen the citations supporting each claim.
  6. P6, Figure 2: In this Figure, DCLK1 is not indicated. I suggest authors put DCLK1 with the interacting proteins on the suitable part. Response: We appreciate the reviewer’s interest in clarifying the role of DCLK1 within the signaling network shown in Figure 2. While DCLK1 has been associated with the regulation of several key pathways depicted (e.g., NF-κB, PI3K/AKT, MAPK, β-catenin), these connections are largely inferred from correlative studies and functional assays, rather than from direct protein–protein interactions or defined pathway mapping. To avoid overstating mechanistic certainty, we have chosen not to insert DCLK1 directly into the figure schematic, which is designed to illustrate core radiation-induced signaling axes in GI injury. Instead, we have clarified DCLK1’s involvement with these pathways in the figure legend and main text, citing relevant studies that support its upstream or modulatory roles. This approach preserves the integrity of the figure while remaining faithful to current evidence.
  7. P6 and 7, Figure 2 legend: Nf-kb and TGF-b; NF-kB and TGF-b. Response: We have corrected these unfortunate formatting errors and thank the reviewer for identifying them.
  8. P7, L233-235 and L262-263, and P8, L270-272: Regarding the reference [99] and others, if DCLK1 mediated signaling system were shown experimentally or even if it were just speculation, that can be illustrated. Response: We thank the reviewer for this helpful suggestion. In the revised manuscript, we have updated Figure 3 to show DCLK1’s suppression of miR-143/145 and its impact on KRAS-driven MAPK signaling.
  9. P8, L273-297: It is interesting that DCLK1 functions as a epithelial generation factor with p53-modulation functions. It would be worth depicting the molecular mechanisms. Response: We thank the reviewer for highlighting the relevance of DCLK1’s modulation of p53-related pathways during epithelial regeneration. While DCLK1 has been implicated in the suppression of p53-driven apoptosis and senescence, especially in cancer and injury models, current evidence supports an indirect regulatory relationship. Specifically, DCLK1 activates upstream pathways such as PI3K/AKT, Wnt/β-catenin, and TGF-β, which are known to intersect with or suppress p53 transcriptional activity. However, a direct physical interaction between DCLK1 and p53 has not been demonstrated. To address the reviewer’s comment, we have clarified this signaling architecture in the revised text. We believe this provides sufficient mechanistic insight while maintaining accuracy regarding the current state of the literature.
  10. P9, Table 2: Appropriate references should be indicated. Response: We added relevant references emphasizing current reviews where possible.
  11. P10, L367-383: This section seems to be not directly associated with DCLK1. Therefore, I recommend authors to edit. Response: We appreciate the reviewer’s attention to maintaining focus on DCLK1. The section in question provides a necessary contextual overview of current therapeutic strategies for radiation-induced damage, including radioprotectors, senolytics, and fibrosis-targeting agents. While these paragraphs do not explicitly discuss DCLK1, they establish the clinical and biological challenges that any novel intervention—including DCLK1-targeted therapies—must address. To maintain logical flow, we retain this section as background and directly introduce DCLK1-specific therapeutic strategies in the subsequent section.
  12. P12, Figure 3: The legend is alright. However, this Figure is hard to understand how DCLK1 plays a role in Radioresistance. Indicate IR-damage not on the arrow but on each cell. Additionally, how do apoptotic cells recover during the process? That should be explained in the legend. Response: Thank you for this insightful critique. We simplified the figure to reflect current gaps in mechanistic understanding and focus on IR’s broad cellular effects. To improve clarity, we revised the legend to pinpoint IR‐induced damage at the cellular level and to state that apoptotic cells do not recover, whereas DCLK1⁺ surviving cells—often spared by quiescence or protective signaling—activate regenerative programs. The updated legend now delineates the divergent fates of irradiated epithelial cells and underscores DCLK1⁺ tuft cells’ role in radioresistance and tissue repopulation.
  13. P13, L481-P14, L577: The description is too long. The only part that contains description on DCLK1 is 5.2.4. Authors had better make best effort to edit the section as concise as possible. Response: We respectfully disagree that the length of the text between P13, L481 and P14, L577 is excessive or tangential. Although only subsection 5.2.4 explicitly names DCLK1, the preceding describes the broader pathway details that are directly relevant to interpreting our subsequent findings, so that readers can fully grasp the novelty and significance of our work. For these reasons, we believe that maintaining the current level of detail is both scientifically justified and necessary for accurate interpretation of DCLK1’s role in autophagy‐linked stress responses.
  14. P16, Table 3: Appropriate references should be indicated. Some of the chemical compounds or drugs, including TEMPOL, should be explained for readers to understand why they were used for the combinatorial treatment for ARS. Response: We respectfully submit that Table 3 was intended purely as a conceptual framework rather than a definitive, literature‐validated treatment regimen. Its purpose is to illustrate a set of hypotheticals, mechanism‐driven interventions for ARS. Because Table 3 is explicitly proposed rather than “current,” requiring individual citations for every entry misunderstands its function. The detailed, evidence‐based rationale for each agent (e.g., TEMPOL’s ROS scavenging, angiotensin-receptor blockade, senolytic clearance, etc.) is already discussed elsewhere in the manuscript where appropriate primary literature is cited. In contrast, Table 3 serves simply as a roadmap of emerging strategies—not as a summary of completed studies. Insisting on references for a table of untested, conceptual combinations would mischaracterize it as a prescriptive protocol rather than the exploratory outline we intended. Therefore, we believe that no additional citations are necessary in Table 3 itself, since each agent’s individual evidence is fully documented in the main text and bibliography.

Round 2

Reviewer 3 Report

Comments and Suggestions for Authors

Although the authors responded to most of my concerns, there are some issues to consider. The authors claimed that “In colon cancer, autophagy inhibition causes p62 to accumulate and co‐localize with DCLK1, preventing DCLK1 degradation, sustaining Nrf2 and Wnt/β-catenin signaling, enhancing antioxidant defenses, and suppressing apoptosis”. However, no evidences and references were provided in the manuscripts. The authors also mentioned that “no study to date directly demonstrates DCLK1 regulating Nrf2‐dependent radioresistance within the tumor microenvironment”. Therefore, there are still lack of the evidences to demonstrate that DCLK1 plays a role in Nrf2-mediated antioxidant responses.  I suggest the authors remove the section 3.1 and “Nrf2” in the table 2 and abstract. 

Author Response

Reviewer Comment:
“Although the authors responded to most of my concerns, there are some issues to consider. The authors claimed that ‘In colon cancer, autophagy inhibition causes p62 to accumulate and co‐localize with DCLK1, preventing DCLK1 degradation, sustaining Nrf2 and Wnt/β-catenin signaling, enhancing antioxidant defenses, and suppressing apoptosis’. However, no evidences and references were provided in the manuscripts. The authors also mentioned that ‘no study to date directly demonstrates DCLK1 regulating Nrf2‐dependent radioresistance within the tumor microenvironment’. Therefore, there are still lack of the evidences to demonstrate that DCLK1 plays a role in Nrf2-mediated antioxidant responses. I suggest the authors remove the section 3.1 and ‘Nrf2’ in the table 2 and abstract.”

Author Response:
We thank the reviewer for the thoughtful critique regarding the mechanistic claims involving DCLK1, autophagy, and Nrf2 signaling. In response, we substantially revised the relevant section (now titled “DCLK1, p62 and Autophagy”) to eliminate any implication of a direct regulatory role for DCLK1 in Nrf2-mediated radioresistance. In section 3.1, we clarify that p62 accumulation under impaired autophagy sequesters Keap1, thereby indirectly stabilizing Nrf2, as supported by prior studies. We removed language suggesting direct DCLK1–Nrf2 regulation and explicitly state: “Although no study to date directly links DCLK1 to Nrf2-driven radioresistance, its stabilization through the p62–autophagy axis contributes to multiple survival pathways and may facilitate oncogenic transformation.” To avoid overinterpretation, we removed Nrf2 from Table 2, revised the abstract to omit reference to antioxidant regulation by DCLK1, and updated Figure 3A to exclude Nrf2 signaling while expanding the depiction of p62-dependent autophagy and DCLK1 stabilization. These revisions better reflect the current evidence and appropriately temper our conclusions.

Reviewer 4 Report

Comments and Suggestions for Authors

Comments to authors:
Authors indicated the molecular structure of the DCLK1 protein and its variants in Figure 1. Moreover, authors have successfully revised the text following my suggestion. This time, I evaluate the revised manuscript acceptable for publication.

Recommendation: Accept in the present form

Author Response

We sincerely thank the reviewer for their constructive feedback throughout the review process. We are pleased that the revisions to Figure 1 and the accompanying text adequately addressed your prior concerns. We appreciate your recommendation for acceptance and your thoughtful evaluation, which helped improve the clarity and rigor of our manuscript.

Round 3

Reviewer 3 Report

Comments and Suggestions for Authors

The authors have addressed my comments. I suggest accept the manuscript for publication.